# A Customizable No-Code Realistic Motion Editor for VRM-Based Avatars

**Po-Hsun Cheng \*** , **Li-Wei Chen and Chia-Hsuan Lin**

Department of Software Engineering and Management, National Kaohsiung Normal University,
Kaohsiung 82444, Taiwan
* Correspondence: cph@mail.nknu.edu.tw

**Abstract:** Avatar actions can be captured using certain gesture sensors or can be predefined by game designers through desktop applications. In other words, developing an online avatar editor could be necessary to specify the detailed actions for use by people who are not game creators. Our research team proposed a web-based toolset, myKLA, to construct and design avatar actions with editor and player features. The goal of myKLA is to help users define the required behaviors of avatars within a set time frame without codes. We used cyber–physical system theory in a software reconstruction initiative. Additionally, an exchangeable JSON file format for predefining the avatar actions was opened and shared here. Furthermore, the cyclomatic complexity of the main code blocks in our toolset was measured and changed using the McCabe approach to fine-tune the performance. An algorithm was proposed for quickly calculating an integrated activity diagram from several sub-activity diagrams. Our research showed that it is easy to create an avatar head and embed it in other web-based applications for additional interaction utilization. Therefore, our findings will be useful in creating and designing new educational tools.

**Keywords:** action definition; code-free; cyber-physical systems; cyclomatic complexity; software design; web-based application





## 1. Introduction

As with most developed countries, Taiwan has deployed national technology education since the summer of 2019. Information education is part of technology education. On the basis of the government education policy, all junior high and senior high schools promote technology-based teaching activities. Courses, textbooks, and educational tools are published for educational usage and even shared on the Internet. Additionally, our research team discovered that most education tools are developed and shareable online for pupils to learn specific knowledge points. However, most educational tools are not funny, vivid, or lively. Such tools may not be engaging for most learners, especially school-age children.

On the other hand, school-age children seem to be more interested in the metaverse's virtual idols. The metaverse has aroused attention in recent years. Several technologies are supported for designing virtual idols, developing virtual scenes, and even writing the required codes to control the virtual idol's actions. However, there is a challenge for most graphic designers to cross from art design to programming design [1,2] For example, Chilana et al. collected feedback from their participants on using MarmalAid and were asked to extend it to other types of applications, such as image editing programs and programming environments [1]. Additionally, Jahanlou et al. mentioned that there are existing efforts in HCI to increase the participation of users and domain experts in various design tasks, such as programming, website creation, and interaction design [2].

Hence, an idea was born to build a bridge to help most visual art workers, even kids, school-age students, and teachers, control their virtual idols to execute predefined actions in a code-free environment. An online knowledge learning assistant, myKLA, was

designed and developed, and its goal is to assist users in quickly defining the required actions of avatars within a specific time duration, as well as promoting the educational tools to share their materials with customizable and animated avatars. This research aimed to design, implement, refactor, and assess a proposed toolset to fill and bridge the gap between graphic designers and web-based programmers.

## 2. Related Works

This section illustrates the following required works, including educational tools, the metaverse, cyber-physical systems (CPSs), and the McCabe method.

### 2.1. Educational Tools

Most advanced countries have included information technology as fundamental education. Further, many technology products have been transformed into education tools in the classroom while new technologies are booming. For example, metaverse game design and information system development can be divided into several phases and engage distinct engineers to design the required tasks for achieving and integrating a complete software product. A full-stack engineer is hard to find and needs a lot of training and learning experience. Even such an engineer has to have artistic talent and logical thinking to leverage all related tasks.

Firstly, several types of research on educational tools are found for art design. For example, in 2020, Terton et al. used three-dimensional (3D) cell shading and two-dimensional (2D) illustrations to provide an engaging and authentic digital learning experience using design-based research principles [3]. Additionally, Han et al. developed two versions of a web-based educational tool, namely, virtual prototyping for e-textile learning. One version utilized a task-oriented approach, and the other used an explorative-oriented method. Students could virtually connect to electronic elements, practice coding, and even view circuit diagrams [4]. Furthermore, Lichty proposed the use of gesture theory and models to improve the creative discipline of students [5]. In 2021, Xing et al. proposed a plot-based virtual reality (VR) education system to provide immersive, explorative, and educational experiences for students. They utilized virtual storytelling technology to promote enthusiasm in user participation [6].

On the other hand, we focus on the programming education research part. In 2019, Cabatuan proposed a Dismath game based on a minimax tree search with an alpha-beta pruning algorithm and showed that the automated gameplay of the Dismath educational tool could be used for logic introduction [7]. In addition, Ruiz et al. proposed a feedback-enriched simulation environment (FENIkS) for learning basic user interface design principles. It is a model-driven educational environment with conceptual domain and presentation models [8].

Additionally, several educational tools have been discovered for other fields. For example, in 2020, Terton et al. proposed a digital educational game, CliNCare, to improve clinical reasoning skills in the delivery of nutrition care to third-year dietetic students [3]. In 2021, Zamani et al. used crossword puzzles as an enjoyable and participatory educational tool accompanied by lectures that could improve management quality in speech therapy sessions [9]. In 2022, Komninos et al. mentioned that CPS constitutes an effective K–12 educational tool. Further, robots and drones have been shown to complement tools of augmented reality and mixed reality concepts [10].

On the basis of the above research outlined [1,2], however, few researchers have paid much attention to the knowledge gaps that still exist between art design and programming design. Most students cannot continuously understand and learn advanced courses because entry-level students cannot enter the introductory course smoothly. Such a situation would affect their further learning pace. To promote learning performance in programming, there is a chance for us to construct a bridge between art design and programming courses. Such a bridge has a couple of characteristics, such as being easy to use, shareable, and customizable, and still having animation capabilities.

## 2.2. Metaverse

In 1995, Bell et al. presented their educational approaches, including multimedia, computerized sessions, and several mute realistic simulations that were complex and expensive to allocate in the classroom [11]. Interestingly, Zhang et al. from China used virtual reality technology in an ideological and political theory course to stimulate students' interest in learning in 2020. Their research outcomes showed that it effectively promoted the understanding of knowledge and the establishment of emotional attitudes and values. It also demonstrated that students might be stimulated by animation-style courses, even if the course is mandatory and likely boring to most students [12].

Further, there is an IEEE standard working group for interfacing cyber and physical worlds, IEEE P2888 [13,14]. It contains six categories, namely, (1) specification of sensor interface for cyber and physical worlds, (2) standard for actuator interface for cyber and physical worlds, (3) standard on the orchestration of digital synchronization between cyber and physical worlds, (4) standard on architecture for virtual reality disaster response training system with six degrees of freedom, (5) evaluation methods of virtual training systems, and (6) holographic visualization for interfacing cyber and physical worlds. A couple of regional and international standards for metaverse technologies are defined in depth and presented shortly.

Fortunately, the VRM consortium proposed and maintained an open-source and platform-independent VRM file format used for 3D models [15]. The VRM file format provides conventional information, including textures and bones. Additionally, a large number of VRM-based avatars that are designed by virtual designers are available on the Internet. Additionally, some similar virtual tutor technologies are available, such as VRoid Studio [16], Wakaru [17], PrprLiv [18], and VRM4U [19]. A comparison of these similar tools is listed in Table 1.

**Table 1.** Comparison of similar software.

| Tool | Advantages | Disadvantages |
| --- | --- | --- |
| VRoid Studio | 1. It is a 3D character creation software.<br>2. It provides a VRoid Hub to share the VRM-based characters that are designed by virtual designers. | 1. It does not provide a tool to self-define the avatar's actions.<br>2. It needs coding skills to demonstrate the avatar's actions. |
| Wakaru | 1. It is a simple and easy-to-use VTuber tool.<br>2. It needs a camera.<br>3. It uses open broadcaster software (OBS) to become a live VTuber immediately.<br>4. It works at a low cost. | 1. It does not provide a tool to self-define the avatar's actions. |
| PrprLive | 1. It is a 2D assistant software.<br>2. It uses OBS with a high frame rate.<br>3. It works at lower computational resource utilization. | 1. It does not support 3D models.<br>2. It does not provide a 3D tool to self-define the avatar's actions.<br>3. It needs coding skills to demonstrate the avatar's actions. |
| VRM4U | 1. It is a runtime VRM loader for Unreal Engine v4.<br>2. It can decorate the avatar. | 1. It needs coding skills to demonstrate the avatar's actions. |

## 2.3. CPS

Since we mention the cyber and physical worlds, this work is related to CPSs. The second edition of Lee and Seshia's book explores how to use finite-state machines (FSMs) to develop embedded systems with the CPS concept [20]. Therefore, using FSMs is one of the core concepts in designing a CPS. In 2019, Chaudhry et al. illustrated that CPS applications are widely used to control the critical infrastructure of various application domains [21]: for example, software [22–24], hardware/software co-design [25,26], big data [27,28], security [21,29], networks [30], vehicles [26,31], industry [32,33], and healthcare [34,35]. These

applications usually collect input data from sensors, estimate the system's current state, and make decisions based on the estimation to automatically control the subsequent processes.

From the software point of view, in 2016, Jiang et al. stated that software and middleware are the souls of CPSs [22,23]. Further, CPSs are an emerging area that cannot work efficiently without proper software handling of the data and business logic [22]. Hence, our adaption has to take care of both the logic and big data views.

From the data point of view, for example, in 2017, Song et al. proposed a big data-driven approach, namely, topic-oriented learning assistance (TOLA), for online learning evolution, which is offered to discover students' learning patterns and guide course improvement [27]. Luo et al. recommended their model and application for big data analytics and proposed a comprehensive framework for integrating it into CPS solutions [28]. Therefore, CPSs were chosen as a good approach for managing big data in this research.

### 2.4. McCabe Method

McCabe proposed a measured approach for software complexity in 1976 [36]. It is so traditional and popular that it has become an indicator of program complexity. The method focuses on the complexity of the control processes. Before system analysis, the researcher has to draw the related activity diagram of the software codes. This software metric approach lets $V(G)$ be the complexity of the control flow, E be the number of edges in the activity diagram, and N be the number of decision points in the activity diagram. Then, the following formula, $V(G) = E - N + 2$, is utilized to calculate the software complexity. This research assessed the software complexity using this approach. McCabe concluded that the higher the cyclomatic complexity, the more prone to errors the system will be. Hence, this theory is suited for measuring the software complexity and related error tendency.

For example, Hutajulu et al. utilized the McCabe method to measure the complexity level of every committed programmer from the beginning of a project to date. They used cyclomatic complexity to decide on the complexity level of a contribution. Additionally, they cited that the McCabe method helped them find quality programmers [37]. In 2020 Chai et al. extracted software structure from code and transformed it using software network theory. Then, they used the McCabe method to analyze the software network and deeply understand the software topology [38]. Based on the above illustration, this research utilized the McCabe method to assess the system's complexity and quality.

### 3. Methods

In the era of advanced technology, many things are gradually being replaced by technology. An intermediate tool, myKLA, was designed and developed as a bridge to cross the gap between art design and the field of programming. Firstly, the avatar development time was reduced by using VRM-based 3D avatars. Secondly, both a virtual designer and programmer used myKLA to replace the writing programs by defining the activities of the avatar. Such a tool can be embedded directly in educational agencies. The following sections include the system design and CPS adoption.

### 3.1. System Design

The myKLA application was designed and implemented with the goal of assisting users in defining the required actions of an avatar within a specific time duration. It contains at least four features: (1) the Editor web page helps users to upload a user-predefined JSON file, select a specific avatar, choose a body part, side, and action, list user-defined actions, demonstrate the user-defined actions online, and download and save user-defined actions in a JSON file; (2) the Player web page is used for uploading a user-predefined JSON file and for showing the user-defined actions online with two avatars; (3) the Move List web page offers a tree-view list for all editable parts of the human, such as the face, limbs, and body; and (4) the Guide web page provides a tutorial for the myKLA application.

Well-known international standards and popular programming languages, such as Hypertext Markup Language (HTML) [39], Cascading Style Sheets (CSS) [40], JavaScript [41],

and JavaScript's related jQuery [42] library, were used to design and develop the web-based application. A remote MariaDB v10.1.48 database stores all required data on the Ubuntu v18.04.6 server. Additionally, the JSON file format was chosen and designed as a data exchanging standard for storing the actions of specific avatars.

The system diagram for myKLA is shown in Figure 1a. Because the designed codes are native and use international standards, the client-side constraints are so flexible that users can browse on most browsers. Therefore, web pages can be shown on a browser that supports HTML5, CSS3, JavaScript, and jQuery. Inside myKLA, the Three.js library is used to create a 3D scene and import the virtual character model. Further, a back-end Node.js server was implemented to effectively support front-end rendering. Additionally, the required and complicated user interface of the editor controls was designed, as shown in Figure 1b. Although the diagrammatic sketch might have minor changes in the final release version, such a prototyped schematic diagram provides a communication path to programmers for further development reference.

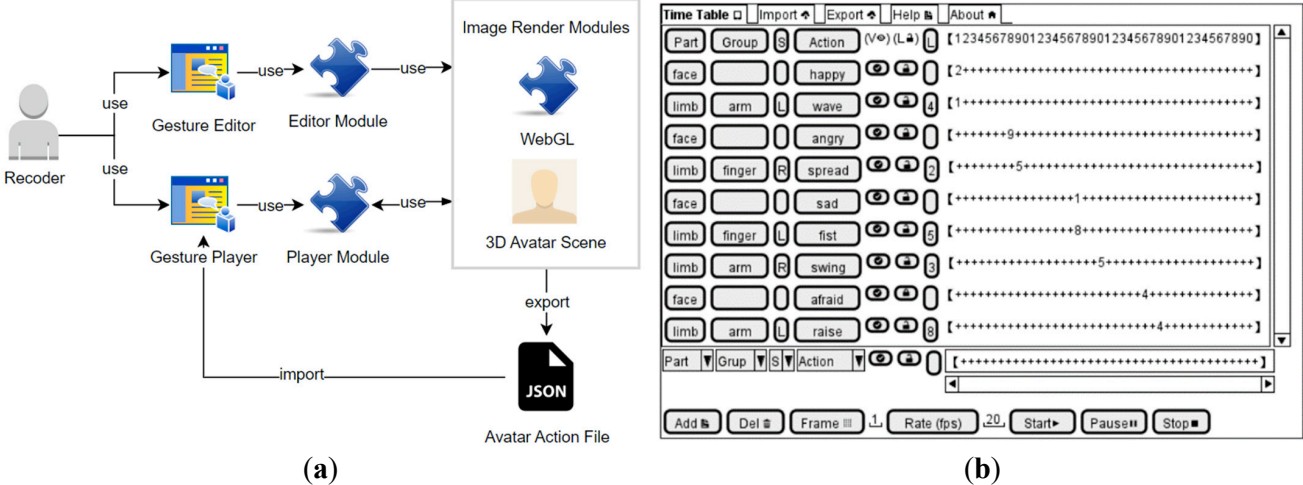

**(a)**                    **(b)**

**Figure 1.** myKLA (**a**) system diagram and (**b**) user interface diagram for editor controls.

### 3.2. CPS Adoption

To simplify the system complexity, CPS theory was adopted to analyze and design the processes using an FSM [20]. That is, a state diagram was used to let hazarded code blocks become easier to understand and decrease the risk of misunderstanding. The primary and complex critical control block was designed for sketching its state diagram, as shown in Figure 2.

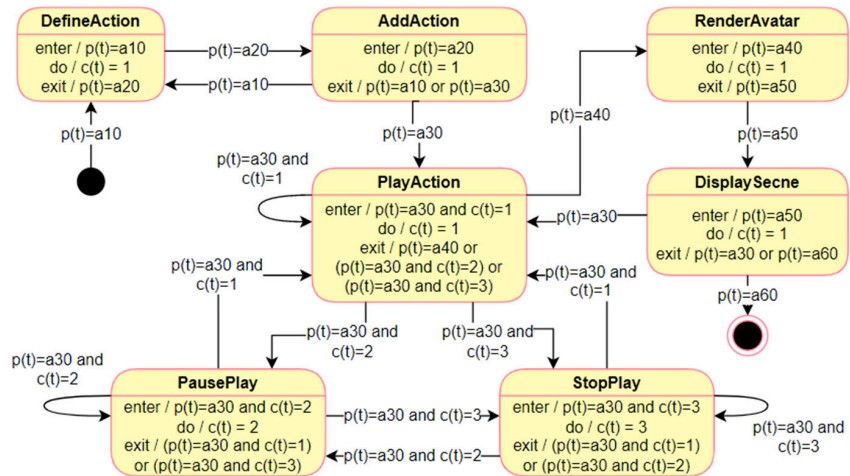

**Figure 2.** State diagram of myKLA.

Let p(t) and c(t) present the program execution status and module selection at time t. When p(t) = a10, the user starts to define an action in the DefineAction state. When p(t) = a20, the user then enters the AddAction state. After completing all action definition processes, the user can enter the PlayAction state and p(t) = a30. If p(t) = a30 and c(t) = 1, then the user enters the PlayAction module. If p(t) = a30 and c(t) = 2, then the user enters the PausePlay module; otherwise, if p(t) = a30 and c(t) = 3, then the user enters the StopPlay module. When p(t) = a30 with a specific c(t) value, the module is still playing, pausing, and stopping the execution cycle and will check the c(t) value iteratively. If p(t) = a40, then the user steps into the RenderAvatar state until p(t) = a50. When p(t) = a50, the avatar is displayed on the browser and exits. Thus, after the state transitions among features were identified, the application was developed by using the derived state diagram.

## 4. Results and Discussion

Our research team spent time designing, developing, implementing, and refactoring the myKLA toolset. It was in production on 4 July 2021. The following subsection illustrates the web-based myKLA toolset, including the user interface design, cyclomatic complexity measurement, performance metrics, and embedded solution.

### 4.1. User Interface Design

In 2018, a question about gender fitting was proposed. Then, a 3D model was utilized to replace the required actors in educational tools. However, we not only wanted to define, design, and implement the actors once but also adopt similar actors in all of the educational tools that we have implemented. After a couple of years of technique collection, skill training, prototype development, method adoption, and process verification, a web-based educational toolset, myKLA, was developed, which provides an environment that lets instructors design and reconfigure the required actor for their educational tool usage. Such a toolset can replace the actor for any type of gender to fit the preference of students.

Generally, at least three male college students were included in the research team and acted as part-time programmers to implement the myKLA web-based application in April 2021. The prototype was constructed in June 2021. In the meantime, three female college students were included in the same team in our laboratory, who provided many comments to improve the design and prepared to revise the next version. For example, Figure 3 shows a screen snapshot of the home page of the proposed myKLA educational toolset, which was designed and beautified by one of the female programmers.

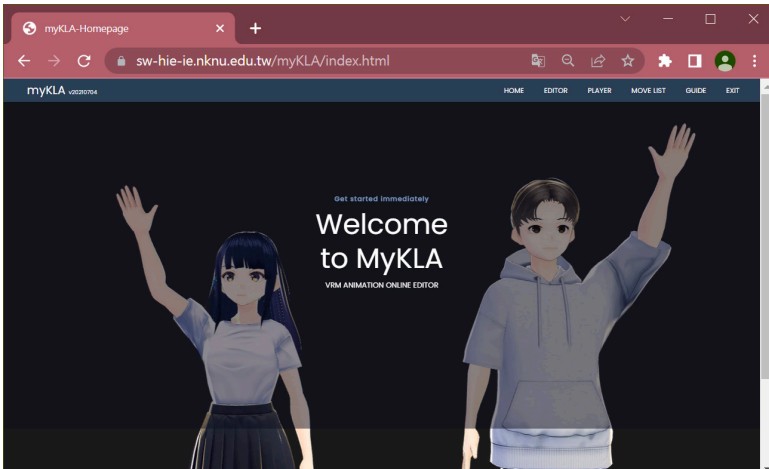

**Figure 3.** myKLA educational toolset home page.

There are two primary functions, namely, the editor and player, in myKLA. As their names suggest, the former is used for editing the detailed actions of specific actors, and the latter can perform the predefined actions for the particular actor. The editor features are

illustrated first. A web page is divided into two panels. In the left panel, there are two 3D models: one female stands on the left, and one male stands on the right. The right panel shows two sections. The upper Edit section selects and defines the action time interval for specific actions.

Inside the Edit section, there are five drop-down lists, including part, group, side, action, and gender from left to right, as well as a sequential identifier for assigning specific actions. Finally, every action accompanies a time interval scroll bar to define the action's playing duration. Note that the default time interval for action play is limited to 60 s, which can be adjusted.

The part drop-down list includes face, limb, body, and location. Further, the group drop-down list is shown as a distinct part selection. For example, it shows three items, such as arm, finger, and leg, after selecting a limb item in the part list. All of the actions are listed in Table 2. The content is also labeled 'Move List' in the menu bar.

**Table 2.** Move list for myKLA.

| Part | Group | Side | Actions |
|---|---|---|---|
| Face | Emotion | Not available | Fun, anger, sorrow, joy, surprise, and blink |
| | Mouth | Not available | A, e, i, o, and u |
| Limb | Arm | Both, left, and right | Hang, swing, lift, hold, wave, stroke, shrug, salute, arm-heart, raise and at-ease |
| | Finger | Both, left, and right | fist, spread, and thumb-up |
| | Leg | Both, left, and right | Walk, left knee, raise the leg, and stretch |
| Body | Waist | Not available | Bow |

After defining a specific action, the action definition will be shown in the lower actions section, as shown in Figure 4a. For example, the boy model can be defined as hanging his left arm for 3 s, swinging his left arm for the next 3 s, and then lifting his left arm for the last 3 s. Meanwhile, the girl model can be determined to lift her right leg in a walking style for 3 s, lift her right knee for the next 3 s, and then raise her right leg for another 3 s. All of the actions inside the move list can be predefined in this editor. Figure 4b demonstrates the player with an imported JSON file exported from the above action editor. Further, Figure 4c is the view of the editor from a small-sized mobile phone.

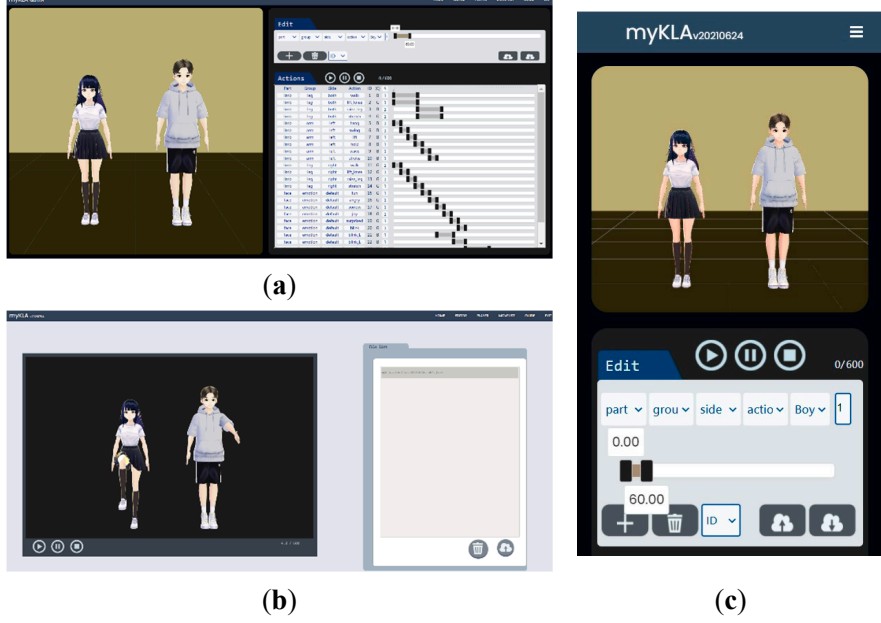

**Figure 4.** Screen snapshot of myKLA: (**a**) editor, (**b**) player, and (**c**) editor in mobile phone.

Additionally, at least five functions are provided to manage the editing process. These functions include adding an action with a plus sign, removing an activity with a trash can, sequential identifier selection using a drop-down list, uploading an external predefined action file with a cloud upload sign, and downloading currently defined actions and saving them in a JSON file format. Table 3 shows the action file syntax of myKLA.

**Table 3.** Action file syntax of myKLA.

```
{
    "actionDefined": [actionSingle, ... ],
    "timeline": [timelineSingle, ... ],
    "moveEndTime": [ ]
}
actionSingle := [
    id: string,
    partNo: number,
    groupNo: number,
    sideNo: number,
    actionNo: number,
    genderNo: number,
    part: string,
    group: string,
    side: string,
    action: string,
    gender: string]
```

Subsequently, the player can be used to perform the actions that are predefined by the JSON files. Figure 5 shows most of the gestures that can be edited by the myKLA editor. For example, Figure 5a–f demonstrates the actions of both limbs, including hanging arms, swinging arms, lifting arms, holding arms, waving arms, stroking arms, shrugging, saluting, heart shape with arms, raising arms, and arms at ease. Of course, the two models demonstrate different actions. Further, Figure 5g,h show that the gestures can be controlled for the left and right limbs by user definition.

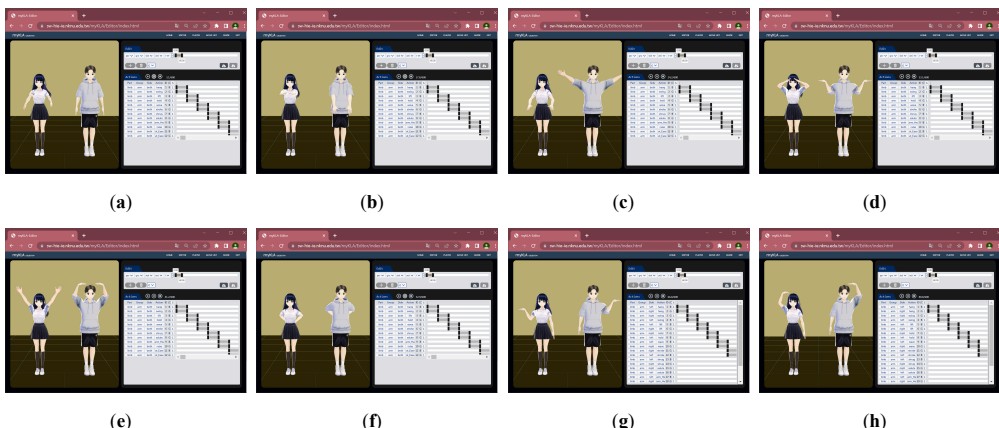

**Figure 5.** Screen snapshot of the myKLA player. (**a**–**f**) demonstrates the actions of both limbs, including hanging arms, swinging arms, lifting arms, holding arms, waving arms, stroking arms, shrugging, saluting, heart shape with arms, raising arms, and arms at ease. (**g**,**h**) show that the gestures can be controlled for the left and right limbs by user definition.

*4.2. Cyclomatic Complexity Measurement*

To minimize the system complexity and promote software performance, the McCabe approach [36,38] was utilized to measure the cyclomatic complexity of major code blocks in myKLA. Firstly, the activity diagram of the main code blocks, such as the Editor web page, was drawn, as shown in Figure 6. The Editor web page shows at least two panels,

such as the Edit panel and the Action panel. There are 33 nodes with 44 paths and 11 nodes with 12 paths for the Edit panel and the Action panel, respectively. Hence, the cyclomatic complexity could be calculated as $44 - 33 + 2 = 13$ and $12 - 11 + 2 = 3$ for the Edit panel and the Action panel, respectively. Although the cyclomatic complexity of the Edit panel is higher, most branches are required for extracting definitions from users.

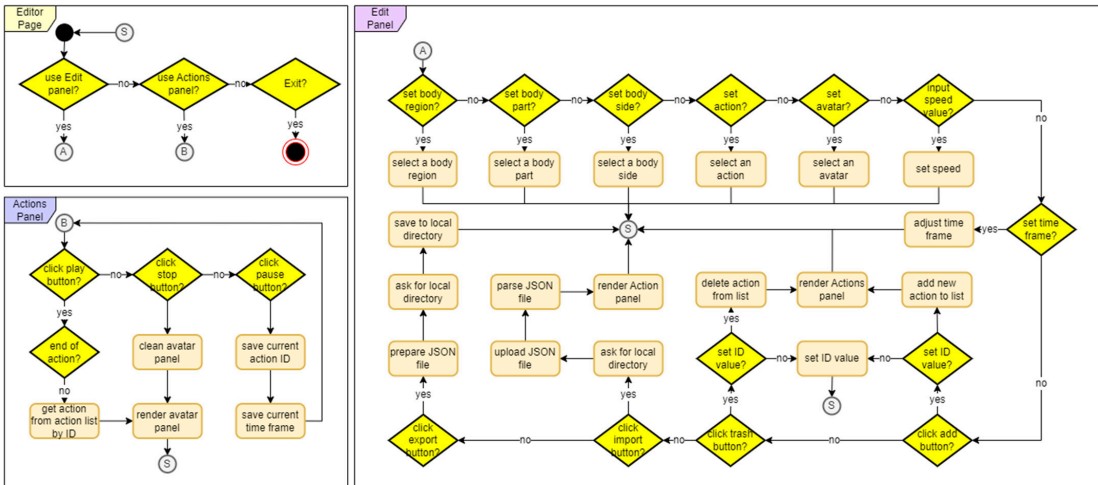

**Figure 6.** Activity diagrams for cyclomatic complexity analysis: Edit panel of the Editor web page.

Additionally, the cyclomatic complexity of the whole Editor web page could be calculated. There are at least two approaches to quickly calculating the nodes and paths of the activity diagram. Firstly, the nodes and paths can be counted directly. Secondly, an algorithm can be derived, as shown in Table 4, to calculate the total nodes and paths of the central activity diagram more efficiently. On the basis of the proposed algorithm, the activity diagram of the Editor web page contains $(33 + 11) + 8 - 3 = 49$ nodes and $(44 + 12) + 7 - 3 = 60$ paths. Hence, there are 49 nodes with 60 paths on the Editor web page, and the cyclomatic complexity of myKLA is $60 - 49 + 2 = 13$, which is the same as the cyclomatic complexity of the Edit panel. The calculated complexity value shows that myKLA contains slightly complex codes with medium testability and moderate risk. If our development team wants to fine-tune the performance of the Editor web page, the Edit panel should be the first choice to concentrate on.

**Table 4.** Algorithm for counting nodes and paths of all subfigures.

```
def getTotalCount()
    # Definition
    SubFig {                              # list of all subfigures
        SF_Name string                   # subfigure: name
        SF_NodeCount integer             # subfigure: node count
        SF_PathCount integer             # subfigure: path count
    }
    MF_NodeCount integer                 # main figure: node count
    MF_PathCount integer                 # main figure: path count
    MF_CrossNodeCount integer            # main figure: cross node count
    T_NodeCount integer                  # total node count
    T_PathCount integer                  # total path count

    # Calculation
    for s in range(SubFig)
        T_NodeCount += s.SF_NodeCount
        T_PathCount += s.SF_PathCount
    T_NodeCount += MF_NodeCount - MF_CrossNodeCount
    T_PathCount += MF_PathCount - MF_CrossNodeCount
```

### 4.3. Performance Metrics

After the implementation of myKLA, system metrics were explored to improve the performance of execution. The system allocated all of the computational components to the distinct Node.js servers at a remote VMware virtual machine under HP ProLiant DL360 Gen9 with two Intel Xeon CPU E5-2620 v4 2.10GHz in eight cores per CPU, 32GB memory, and a 500GB hard disk.

Firstly, Table 5 shows the last performance metrics of myKLA, which were measured on 2 September 2022. The client environment of our metrics was an Intel Core i7-8565U with 16GB memory, an Intel UHD Graphics 620 video card, and Google Chrome v106.0.5249.119 (64-bit). Both the Editor and Player pages showed boy and girl avatars with file sizes of approximately 10.0 MB and 14.2 MB, respectively. On the Editor web page, the average time expenses of our codes and the third-party libraries were 5858 ms (213 KB) and 1.107 ms (2591 KB), respectively. It makes sense that our codes call several libraries and wait for completion; however, our codes have a lot of optimization opportunities in terms of improving the execution performance. Additionally, this shows that loading the male avatar (10.312 MB) and female avatar (14.662 MB) takes approximately 4488 ms and 4266 ms, respectively. Thus, it is necessary to use a model with as small a file size as possible to reduce the average downloading time from the remote server.

**Table 5.** Performance metrics.

| Features | Activities | File Size (KB) | Average Time Expense (ms) |
|---|---|---|---|
| Editor | Initialization: load our codes | 213 | 5858 |
| | Initialization: load the third-party libraries | 2591 | 4618 |
| | Initialization: load boy avatar | 10,312 | 4488 |
| | Initialization: load girl avatar | 14,662 | 4266 |
| Player | Initialization: load our codes | 136 | 1107 |
| | Initialization: load the third-party libraries | 2267 | 779 |
| | Initialization: load boy avatar | 10,312 | 4176 |
| | Initialization: load girl avatar | 14,662 | 3607 |
| | If the user tries to use one JSON file | 13 | 12 |

On the Player web page, the average time expenses of our codes and the third-party libraries were 1.107 ms (136 MB) and 779 ms (2267 KB), respectively. Additionally, our codes need further optimization. For the system to load the same pair of avatars, it takes approximately 4176 ms and 3607 ms, respectively. Alternatively, the JSON file size is so small, at 13.1 KB, that its average time expense, 12 ms, is negligible.

### 4.4. Embedded Solution

Within the COVID-19 disease period, due to the impact of the epidemic, many people began to use online environments and systems, but this also reduced the opportunity to communicate with others. If virtual characters are used in the system, we can not only achieve the effect of companionship and increase interactivity but also relatively improve users' willingness to use the system. Therefore, users can use myKLA to establish virtual character action files so that virtual characters can assist in the role of companionship.

As the system can be opened anytime and anywhere, it increases convenience and improves efficiency. For example, a male avatar was created with a head component by using the myKLA intermediate tool. Then, such a head component was embedded in the bottom-left corner of another educational tool, myBlock2, which is used for Scratch and Python programming training, and the avatar squinted, blinked, and opened his eyes, as shown in Figure 7a–c, respectively.

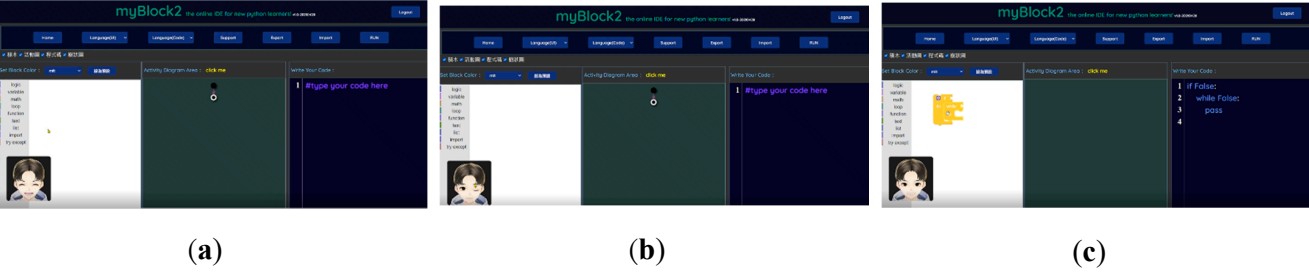

<div align="center">(<b>a</b>)       (<b>b</b>)       (<b>c</b>)</div>

**Figure 7.** Embedded myKLA avatar component in the myBlock2 tool. Python programming training, and the avatar squinted (**a**), blinked (**b**), and opened his eyes (**c**).

This research utilized the method of real-time detection of interactive computing between the front-end web page and the back-end server to avoid the computational delay due to the rendering of the 3D scene on the web page. It can effectively provide a real-time preview function of scenes that require a large number of renderings, which is convenient for users to confirm the recognition effect. Hence, external factors such as the lens angle can be adjusted in real time.

Additionally, by using JSON files as the storage format, future avatar system developers are able to apply their output files with greater scalability. Practitioners can select a different avatar to accompany the challenges based on their preferences because the avatar is not stored in the output JSON file. This proposed technology lowers the barrier to entry for specifying the specific action of a virtual character, capturing those activities in a JSON file, and playing back the avatar actions. Users may manage the system to carry out their ideas with little effort because it is so intuitive and friendly. This flexibility allows users to be more ready to take risks and lessen their sense of learned helplessness.

However, the myKLA tool has some limitations, including the following: (1) exploiting web browsers as target platforms; (2) integrating with the JavaScript interface; (3) employing a predefined JSON file format to record and exchange activities among the player and editor; (4) adopting the VRM file format for avatar models; (5) confining avatar activities with provided features, as shown in Table 1; and (6) using 60 s as the default time interval for activity playing. Certainly, these constraints can be resolved by further extending more features as needed.

For a graphic designer without web programming skills, myKLA can be used to define an avatar's activities without the requirement of coding. On the other hand, if a web programmer is without avatar design capability, myKLA provides a lightweight user interface to handle the avatar directly. Hence, myKLA acts as a type of bridge to compensate for the gap between virtual art designers and web programmers.

## 5. Conclusions

Most advanced countries have listed information technology as fundamental education. However, not many tools can be reconfigured to act as a bridge between art design and programming educational tools. Therefore, CPS theory was adopted to implement a toolset, myKLA, to design and record the required actions for a selected avatar without programming skills. Additionally, the actor model can be designed by users that do not have programming skills or can be directly downloaded from certain VRM-based repositories, such as VRoid Hub, which can be found on the Internet. In addition, an exchangeable JSON file format for predefining the avatar actions was opened and shared here, and an algorithm was proposed for quickly calculating an integrated activity diagram from several sub-activity diagrams. The proposed toolset can be embedded easily into educational tools and act as a learning assistant. The customizability and absence of code in the proposed toolset contribute to the sustainability of software usage. Furthermore, we utilized McCabe's complexity metric to reduce the complexity of the toolset to increase its development and maintenance sustainability.

**Author Contributions:** Conceptualization, P.-H.C. and C.-H.L.; methodology, P.-H.C., L.-W.C. and C.-H.L.; software, C.-H.L.; validation, P.-H.C., L.-W.C. and C.-H.L.; formal analysis, P.-H.C. and L.-W.C.; investigation, P.-H.C.; resources, P.-H.C.; data curation, P.-H.C. and C.-H.L.; writing—original draft preparation, P.-H.C.; writing—review and editing, P.-H.C. and L.-W.C.; visualization, P.-H.C.; supervision, P.-H.C.; project administration, P.-H.C.; funding acquisition, P.-H.C. All authors have read and agreed to the published version of the manuscript.

**Funding:** This research was funded by the 2021 Ministry of Science and Technology, Taiwan grant number MOST 110-2629-E-017-001.

**Institutional Review Board Statement:** This study does not contain identifying information, so it did not require ethical approval. No animal studies are presented in this manuscript. No potentially identifiable human images or data is presented in this study.

**Informed Consent Statement:** Not applicable.

**Data Availability Statement:** The datasets used and/or analyzed during the current study are available from the corresponding author upon reasonable request.

**Acknowledgments:** The authors express their gratitude for all of the participants from the Department of Software Engineering and Management (SEM) and the Information Education Center (IEC) of the National Kaohsiung Normal University, Kaohsiung, Taiwan.

**Conflicts of Interest:** The authors declare no conflict of interest.

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
