# Peer review of "A Customizable No-Code Realistic Motion Editor for VRM-Based Avatars"

_sustainability, doi:10.3390/su15021182_

Round 1

Reviewer 1 Report

This paper is a documentation of the software development process. In this paper, the authors describe a program-less Avator online editor, construct a toolset through CPS theory, and evaluate the complexity and quality of the system using McCabe. In the case of myKLA, the functionality is generally as described in the text.

Essentially, this paper is too long and boring in describing the operational context of myKLA, and there are many self-perceived descriptions, such as the one in [line 209-211], that lack its basis and validation. And these descriptions are often used without a clear and sufficient motivation. The title is "VRM-Based", but the article does not mention a single word about it.

This paper has a potential to be accepted, but some important points must be clarified or amended before proceeding and taking positive action.

1.          The abstract states that the goal of myKLA is to help users define the desired behavior of an avatar within a set time frame without a code. However, the authors do not seem to have included similar software available in this particular area and compared it with them. Examples include the following: VRoid Studio, PrprLiv, wakaru, and other free software with similar functionality can be combined with OBS to put Avator into teleconferencing, teaching, or streaming live, and the teaching resources are abundant, the setup threshold is low, and it is already used by a significant number of students during remote classes. The VRM4U on GitHub has more sophisticated features and greater scalability, with automatic skeleton generation and Rig control. Therefore, there may be a need to supplement myKLA's functionality or other application development advantages.

2.          In this paper, we keep emphasizing "building a bridge between art design and programming" [Line 94-96, 160-161, 356], but if we use myKLA as the bridge, it seems to be just a web-based 3D character tool created by programmers with CPS theory, which is almost the same as developing other application software. It seems to have little relevance to "crossing the field of art design and programming".

3.          In “4.3. Performance Metrics” [line 301- 316], the text of the statement does not present the performance improvements of myKLA.

4.          The actor model can be designed by the users themselves or directly download from the Internet. [Line 358],No descriptions or applications of related functions were found in the myKLA software, please add them.

Author Response

Response to Reviewer 1 Comments

Point 1: This paper is a documentation of the software development process. In this paper, the authors describe a program-less Avator online editor, construct a toolset through CPS theory, and evaluate the complexity and quality of the system using McCabe. In the case of myKLA, the functionality is generally as described in the text.

Response 1: Thanks for your highlights.

Point 2: Essentially, this paper is too long and boring in describing the operational context of myKLA, and there are many self-perceived descriptions, such as the one in [lines 209-211], that lack basis and validation. And these descriptions are often used without clear and sufficient motivation. The title is "VRM-Based", but the article does not mention a single word about it.

Response 2: (1) We revised the original [lines 209-211] description with a more objective mood on [lines 223-224]. (2) We append a paragraph to mention the role and utilization of the VRM-based avatar in our research on [lines 119-126] in the [2.2. Metaverse] section, and on [lines 172-175] in the [3. Methods] section.

Point 3: This paper has the potential to be accepted, but some important points must be clarified or amended before proceeding and taking positive action.

Response 3: Thanks for your highlights.

Point 4: The abstract states that the goal of myKLA is to help users define the desired behavior of an avatar within a set time frame without a code. However, the authors do not seem to have included similar software available in this particular area and compared it with them. Examples include the following: VRoid Studio, PrprLiv, wakaru, and other free software with similar functionality can be combined with OBS to put avatar into teleconferencing, teaching, or streaming live, and the teaching resources are abundant, the setup threshold is low, and it is already used by a significant number of students during remote classes. The VRM4U on GitHub has more sophisticated features and greater scalability, with automatic skeleton generation and Rig control. Therefore, there may be a need to supplement myKLA's functionality or other application development advantages.

Response 4: We add a paragraph and Table 1 to illustrate and compare similar software on [lines 119-126] in the [2.2. Metaverse] section.

Point 5: In this paper, we keep emphasizing "building a bridge between art design and programming" [Line 94-96, 160-161, 356], but if we use myKLA as the bridge, it seems to be just a web-based 3D character tool created by programmers with CPS theory, which is almost the same as developing other application software. It seems to have little relevance to "crossing the field of art design and programming".

Response 5: We add some sentences at [line 172-175] in the [3. Methods] section to reflect this comment.

Point 6: In “4.3. Performance Metrics” [line 301- 316], the text of the statement does not present the performance improvements of myKLA.

Response 6: We revised the paragraph at [line 328-332] in the [4.3. Performance Metrics] section.

Point 7: The actor model can be designed by the users themselves or directly download from the Internet. [Line 358],No descriptions or applications of related functions were found in the myKLA software, please add them.

Response 7: We revised the sentence at [lines 387-388] in the [5. Conclusion] section. Also, the related VRM-based repository was mentioned in [lines 119-126] and Table 1 in [2.2. Metaverse] section.

Reviewer 2 Report

Introduction

Not a single citation was made in the inroduction section. Categorical statements were made without any support citation. The objective - detailing the main objective and specific objectives were not clearly stated. The section failed to identify the gap with supporting citations indicating where such gaps were identified.  These corrections should be affected to show why this study is worth doing and improve the quality of the paper. 

 Lines 27 -28: Recast the sentences to reflect meaningful position. 

Related works

Line 68:  Guseinova, et al., (cite properly)

Line 74" Chakraverty, et al. (cite properly)

Most of the citations in this section used the wrong format. Revisit all the citations. 

Line 174: Web development technologies and tools 

The number of citations in the related works should be improved. 

using the words, "tried" indicates that the researchers did not complete the work or made an inconclusive efforts that did not lead to completion of the study. Tried has been used in many places in this work. 

Generally, the use of pronouns (we, our, I, you) should be avoided. 

Line 297: Discuss the complexity value of 13 as it relates to myKLA.  

Line 359: In the conclusion, not having a programming skill is stated as a unique advantage of MyKLA, but available avatars suggest that a user does not require programming skills to use the product.  Provide clarity on the statement. 

Line 363: In this case state emphatically that the developed toolset could be embbeded easily in eductional tools..... (the use of hope indciates that the work has not been done and researchers are not sure of its applicability)

There is lack of clarity in the outcome of this study, which should have been used to compare with other avatar editors already in the open domain. 

As a user, why should I opt for myKLA? 

Generally, the grammar is poor and should be improved. 

Author Response

Response to Reviewer 2 Comments

Point 1: Not a single citation was made in the introduction section. Categorical statements were made without any support citation. The objective - detailing the main objective and specific objectives were not clearly stated. The section failed to identify the gap with supporting citations indicating where such gaps were identified. These corrections should be affected to show why this study is worth doing and improve the quality of the paper.

Response 1: Thanks for your highlights. We revised [lines 38-45] in the [1. Introduction] section.

Point 2: Lines 27 -28: Recast the sentences to reflect meaningful position.

Response 2: We revised [lines 27-29] in the [1. Introduction] section.

Point 3: Related works: Line 68: Guseinova, et al., (cite properly)

Response 3: We removed it.

Point 4: Related works: Line 74" Chakraverty, et al. (cite properly)

Response 4: We removed it.

Point 5: Related works: Most of the citations in this section used the wrong format. Revisit all the citations.

Response 5: We revised all citations.

Point 6: Line 174: Web development technologies and tools. The number of citations in the related works should be improved.

Response 6: We revised [lines187-192] in the [3.1. System Design] section.

Point 7: Using the words, "tried" indicates that the researchers did not complete the work or made an inconclusive efforts that did not lead to completion of the study. Tried has been used in many places in this work.

Response 7: We replaced the “tried” word with an appropriate word at [lines 171, 231, and 384-385].

Point 8: Generally, the use of pronouns (we, our, I, you) should be avoided.

Response 8: We revised the pronouns with other appropriate terms at [lines 19, 46, 66, 85, 155-156, 171, 178, 191-192, 193, 197, 199, 207-208, 210, 214, 231, 233, 236, 245, 251-252, 260, 263-264, 267, 269, 276, 292-294, 304-305, 348, 361, 367, 384-385, and 391].

Point 9: Line 297: Discuss the complexity value of 13 as it relates to myKLA.

Response 9: We revised [lines 310-311] in the [4.2. Loop Complexity Measurement] section.

Point 10: Line 359: In the conclusion, not having a programming skill is stated as a unique advantage of MyKLA, but available avatars suggest that a user does not require programming skills to use the product. Provide clarity on the statement.

Response 10: We revised [lines 387-388] in the [5. Conclusion] section.

Point 11: Line 363: In this case state emphatically that the developed toolset could be embbeded easily in eductional tools..... (the use of hope indciates that the work has not been done and researchers are not sure of its applicability)

Response 11: We revised [lines 387-388] in the [5. Conclusion] section.

Point 12: There is lack of clarity in the outcome of this study, which should have been used to compare with other avatar editors already in the open domain.

Response 12: We added a paragraph and Table 1 to compare several related software on [lines 119-126] in the [2.2 Metaverse] section.

Point 13: As a user, why should I opt for myKLA?

Response 13: We added a paragraph on [lines 369-380] in the [4.4. Embedded Solution] section.

Point 14: Generally, the grammar is poor and should be improved.

Response 14: Grammar has been improved as much as possible by using Grammarly tool.

Reviewer 3 Report

The current study presented myKLA, a web-based toolset for creating and designing avatar actions with editor and player characteristics. MyKLA's purpose is to assist users in establishing the desired behaviors of avatars within a set time frame without the employment of codes and programming skills. The research showed that the employment of an avatar head embedded in other web-based applications could enhance interaction. The authors concluded by highlighting the importance of their findings in designing future educational tools.

The topic is interesting. Indeed, there is an urgent need for new tools that bridge the gap between art design and the programming fields.  Many teachers, students, or researchers who lack programming skills want to engage virtual idols in their programs. The paper has the potential to advance the current knowledge. The researchers describe their tool in detail. The manuscript is quite clear. Figures are useful.

I would like to make some minor suggestions:

-In the introduction, the authors suitably highlight the existing research gap as well as the importance of their contribution. However, the introduction presents some data without citing references to relevant research. Given that introduction should review key publications, I would suggest citing some representative studies that confirm what is mentioned in the introduction. The same applies to the second section (lines 90-95). It would be better to support this argument with relevant research.

-It would be necessary to clarify the type of this research from the beginning

-It would be useful to mention some limitations of your study

- Check also formatting, spelling, English language, and the reference style (i.e line 127. Sometimes the authors repeat the same verbs in the same paragraph i.e proposed lines 132-138. Regarding table 1, it would be better to use the journal’s template).

Author Response

Response to Reviewer 3 Comments

Point 1: The current study presented myKLA, a web-based toolset for creating and designing avatar actions with editor and player characteristics. MyKLA's purpose is to assist users in establishing the desired behaviors of avatars within a set time frame without the employment of codes and programming skills. The research showed that the employment of an avatar head embedded in other web-based applications could enhance interaction. The authors concluded by highlighting the importance of their findings in designing future educational tools.

Response 1: Thanks for your highlights.

Point 2: The topic is interesting. Indeed, there is an urgent need for new tools that bridge the gap between art design and the programming fields. Many teachers, students, or researchers who lack programming skills want to engage virtual idols in their programs. The paper has the potential to advance the current knowledge. The researchers describe their tool in detail. The manuscript is quite clear. Figures are useful.

Response 2: Thanks for your highlights.

Point 3: I would like to make some minor suggestions: In the introduction, the authors suitably highlight the existing research gap as well as the importance of their contribution. However, the introduction presents some data without citing references to relevant research. Given that introduction should review key publications, I would suggest citing some representative studies that confirm what is mentioned in the introduction. The same applies to the second section (lines 90-95). It would be better to support this argument with relevant research.

Response 3: Thanks for your highlights. Further, we cited [34-35] to support our argument on [lines 38-45] in the [1. Introduction] section and on [line 93] in the [2. Related Works] section.

Point 4: It would be necessary to clarify the type of this research from the beginning

Response 4: We append one sentence to illustrate our research type on [lines 51-53] in the [1. Introduction] section.

Point 5: It would be useful to mention some limitations of your study.

Response 5: We append a paragraph to illustrate the limitations of our study on [lines 369-380] at the end of [4.4. Embedded Solution] section.

Point 6: Check also formatting, spelling, English language, and the reference style (i.e line 127. Sometimes the authors repeat the same verbs in the same paragraph i.e proposed lines 132-138. Regarding table 1, it would be better to use the journal’s template).

Response 6: (1) We revised all formatting, spelling, English language, and reference style with recommended ACS style guide. (2) The same verb in the same paragraph problem was also revised. (3) Table 1 and 4 were formatted using the journal’s table template.

Round 2

Reviewer 1 Report

The study has made a considerable contribution. The methods used in the study are appropriate, novel, and lead to clear results. The conclusion is brief and appropriate. Other suggested aspects are covered in the revised version. The study and its interpretation are supported by many references.

Reviewer 2 Report

Lines 19-21

Further, the research showed that it is easy to create an avatar head and embed it in other web-based applications for additional interaction utilization. Therefore, the findings suggest that it would be useful in creating and designing new educational tools.

line 29: all junior and high school .........

line 40: recast the sentence

line 47: an idea aroused

line 94: also, several educational tools were .......

lines 512-520: references did not follow the standard format.

this work requires further English editing. 

NOTE: the use of Grammarly does not ensure the right grammar and structure. It provides suggestions. 
